# Increased Mortality in *SDHB* but Not in *SDHD* Pathogenic Variant Carriers

**DOI:** 10.3390/cancers11010103

**Published:** 2019-01-17

**Authors:** Johannes A. Rijken, Leonie T. van Hulsteijn, Olaf M. Dekkers, Nicolasine D. Niemeijer, C. René Leemans, Karin Eijkelenkamp, Anouk N.A. van der Horst-Schrivers, Michiel N. Kerstens, Anouk van Berkel, Henri J.L.M. Timmers, Henricus P.M. Kunst, Peter H.L.T. Bisschop, Koen M.A. Dreijerink, Marieke F. van Dooren, Frederik J. Hes, Jeroen C. Jansen, Eleonora P.M. Corssmit, Erik F. Hensen

**Affiliations:** 1Department of Otolaryngology/Head and Neck Surgery, Amsterdam UMC, Vrije Universiteit Amsterdam, De Boelelaan 1117, 1081 HZ Amsterdam, The Netherlands; cr.leemans@vumc.nl (C.R.L.); E.F.Hensen@lumc.nl (E.F.H.); 2Department of Endocrinology and Metabolic Diseases, Leiden University Medical Center, 2333 ZA Leiden, The Netherlands; lvanhulsteijn@hotmail.com (L.T.v.H.); O.M.Dekkers@lumc.nl (O.M.D.); nniemeijer@ysl.nl (N.D.N.); E.P.M.van_der_Kleij-Corssmit@lumc.nl (E.P.M.C.); 3Departments of Epidemiology, Leiden University Medical Center, 2333 ZA Leiden, The Netherlands; 4Department of Endocrinology, University Medical Center Groningen, University of Groningen, Hanzeplein 1, 9713 GZ Groningen, The Netherlands; k.eijkelenkamp@umcg.nl (K.E.); a.n.a.van.der.horst@umcg.nl (A.N.A.v.d.H.-S.); m.n.kerstens@umcg.nl (M.N.K.); 5Division of Endocrinology, Department of Internal Medicine, Radboud University Medical Center, Geert Grooteplein Zuid 10, 6525 GA Nijmegen, The Netherlands; anouk.vanberkel@radboudumc.nl (A.v.B.); henri.timmers@radboudumc.nl (H.J.L.M.T.); 6Department of Otolaryngology/Head and Neck Surgery, Radboud University Medical Center, Geert Grooteplein Zuid 10, 6525 GA Nijmegen, The Netherlands; dirk.kunst@radboudumc.nl; 7Department of Endocrinology and Metabolism, Amsterdam UMC, University of Amsterdam, Meibergdreef 9, 1105 AZ Amsterdam, The Netherlands; p.h.bisschop@amc.uva.nl; 8Department of Endocrinology, University Medical Centre Utrecht, Heidelberglaan 100, 3584 CX Utrecht, The Netherlands; k.dreijerink@vumc.nl; 9Department of Endocrinology, Amsterdam UMC, Vrije Universiteit Amsterdam, De Boelelaan 1117, 1081 HZ Amsterdam, The Netherlands; 10Department of Clinical Genetics, Erasmus MC, University Medical Center Rotterdam, Doctor Molewaterplein 40, 3015 GD Rotterdam, The Netherlands; m.vandooren@erasmusmc.nl; 11Department of Clinical Genetics, Leiden University Medical Center, De Boelelaan 1117, 1081 HZ Leiden, The Netherlands; F.J.Hes@lumc.nl; 12Department of Otolaryngology/Head and Neck Surgery, Leiden University Medical Center, Albinusdreef 2, 2333 ZA Leiden, The Netherlands; j.c.jansen@lumc.nl

**Keywords:** *SDHB*, *SDHD*, mortality, paraganglioma, pheochromocytoma

## Abstract

Germline mutations in succinate dehydrogenase subunit B and D (*SDHB* and *SDHD*) are predisposed to hereditary paraganglioma (PGL) and pheochromocytoma (PHEO). The phenotype of pathogenic variants varies according to the causative gene. In this retrospective study, we estimate the mortality of a nationwide cohort of *SDHB* variant carriers and that of a large cohort of *SDHD* variant carriers and compare it to the mortality of a matched cohort of the general Dutch population. A total of 192 *SDHB* variant carriers and 232 *SDHD* variant carriers were included in this study. The Standard Mortality Ratio (SMR) for *SDHB* mutation carriers was 1.89, increasing to 2.88 in carriers affected by PGL. For *SDHD* variant carriers the SMR was 0.93 and 1.06 in affected carriers. Compared to the general population, mortality seems to be increased in *SDHB* variant carriers, especially in those affected by PGL. In *SDHD* variant carriers, the mortality is comparable to that of the general Dutch population, even if they are affected by PGL. This insight emphasizes the significance of DNA-testing in all PGL and PHEO patients, since different clinical risks may warrant gene-specific management strategies.

## 1. Introduction

Paragangliomas (PGL) are rare tumors that originate from cells of neural crest origin in the paraganglia associated with the autonomic nervous system. PGL can be subdivided into head and neck paragangliomas (HNPGL), pheochromocytomas (PHEO), and thoracic and abdominal extra-adrenal PGL (sympathetic PGL; sPGL). An increasing number of genes are associated with hereditary PGL/PHEO. Most frequently, hereditary PGL syndrome is caused by genes encoding subunits or cofactors of succinate dehydrogenase (SDH), such as *SDHA/B/C/D/AF2*. Other associated genes are *RET, NF1, VHL, HIF2A, FH, TMEM127*, and *MAX* [1,2]. In the Netherlands, pathogenic variants in *SDHD* are the most prevalent cause of PGL syndrome, followed by variants in *SDHB* and *SDHA* [3,4]. Although all SDHx genes encode subunits of the same SDH complex and pathogenic variants all disrupt its enzymatic function, different genes are associated with different phenotypes. The reported lifelong penetrance of pathogenic *SDHB* variants (22–42%) [5,6] is considerably lower than the penetrance of paternally inherited *SDHD* mutations (88–100%) [7,8,9,10]. 

When pathogenic *SDHB* variants cause disease, the clinical outcome is reported to be less favorable than that in *SDHD*-linked disease. *SDHB* mutation carriers are reported to develop metastatic PGL more frequently and patients with metastatic disease associated with *SDHB* variants are reported to have a poor 5-year survival rate compared to patients with metastatic disease associated with other causative genes [11]. The mortality of *SDHB* variant carriers is currently unknown [12]. In this study we estimate the mortality for a nationwide cohort of *SDHB* variant carriers and compare this risk with the mortality of *SDHD* variant carriers and that of the general Dutch population. 

## 2. Subjects and Methods

### 2.1. Eligibility Criteria

The cohort of pathogenic germline variant carriers (hereafter variants) in *SDHB* included in this study has been described in detail previously [6,13]. The mortality of this nationwide *SDHB*-linked cohort was compared with the mortality of the general Dutch population and with the mortality of an updated cohort of *SDHD* variant carriers, which has been described previously [12]. Only *SDHD* variant carriers with paternal inheritance were included. Carriers of *SDHD* variants were identified using the database of the Laboratory for Diagnostic Genome Analysis (LDGA) at the Leiden University Medical Center (LUMC), a tertiary referral center for patients with PGL. Screening for *SDH* variants was performed in all persons diagnosed with PGL who agreed to genetic testing. 

Screening for *SDHB* and *SDHD* variants was performed by direct sequencing of peripheral blood leucocytes using the Sanger method on an ABI 377 Genetic Analyzer (Applied Biosystems, Carlsbad, California) and by multiplex ligation-dependent probe amplification (MLPA) using the P226 MLPA kit (MRC Holland, Amsterdam, the Netherlands). Family members of index patients were tested for the family-specific variant. All variants described in this study were submitted to the Leiden Open (source) Variation Database LOVD database (http://chromium.liacs.nl/lovd_sdh). *SDHB* and *SDHD* germline variants were classified according to the international guidelines put forth by Plon et al. [14]. *SDHD* variants were described using the reference sequence NG_012340.1 covering SDHB transcript NM_003000.2, and NG_012337.1 covering SDHD transcript NM_003002.2, available from the TCA Cycle Gene Variant Database LOVD database. In this manuscript we report pathogenic or likely pathogenic variants, including missense mutations in highly conserved regions that are determined to be likely pathogenic as germline mutations based partly on mutation prediction analyses. Information on amino acid conservation can be found in the LOVD database (http://chromium.liacs.nl/lovd_sdh). Further information including mutation prediction analyses can be obtained on request. 

The study was approved by the Medical Ethics Committee of the Leiden University Medical Center; participating centers complied with their local Medical Ethics Committee requirements. Written informed consent was obtained from the parents/guardians of individuals under 18 years of age. 

### 2.2. Clinical Characteristics

Clinical data were retrieved from medical records. Pathogenic variant carriers were investigated for occurrences of PGL and/or PHEO according to the structured protocols used for standard care in the Netherlands for PGL or PHEO patients [15,16]. Patients were offered clinical surveillance for PGL/PHEO at the departments of otorhinolaryngology and endocrinology. For asymptomatic *SDHB* and *SDHD* variant carriers older than 18 years of age, surveillance consisted of magnetic resonance imaging (MRI) of the head and neck region once every 2–3 years, and MRI or computed tomography (CT) scans of the thorax, abdomen, and pelvis once every 1–2 years in *SDHB* variant carriers. Biochemical screening was performed annually on *SDHB* variant carriers, and every 1–2 years on *SDHD* variant carriers. This screening measured levels of (nor)epinephrine, vanillylmandelic acid, dopamine, (nor)metanephrine, and/or 3-methoxytyramine in two 24-hour urinary samples (depending on the Academic Center in which urinary measurement(s) were performed), and/or plasma free (nor)metanephrine and 3-methoxytyramine. In cases of excessive catecholamine secretion (i.e., any value above the upper reference limit), radiological assessment by MRI or CT scans of the thorax, abdomen, and pelvis, and/or 123I metaiodobenzylguanidine (MIBG) scans, positron emission tomography with 2-deoxy-2-[fluorine-18]fluoro-D-glucose (18F-FDG PET) scans, 18F-L-dihydroxyphenylalanine (18F-DOPA) PET-scans, or positron emission tomography with 1,4,7,10-tetraazacyclododecane-NI, NII, NIII, NIIII-tetraacetic acid (D)-Phe1-thy3-octreotide (68Ga-DOTATOC PET) scans were performed to identify potential sources of excessive catecholamine production. In cases without available tumor histology, tumors were classified as paraganglionic based on their specific characteristics in CT and/or MRI scans. When in doubt, additional nuclear medicine imaging studies were performed in order to confirm the diagnosis. At the time of this study, there were no national, structured protocols for surveillance in *SDHB* mutation carriers younger than 18 years of age. Therefore, the method and interval of surveillance in this age category varied between centers. 

In case of a diagnosis of HNPGL, PHEO or sPGL, intensified surveillance or treatment was offered. Surgical resection was generally the preferred treatment option for PHEO or sPGL. In cases of HNPGL, the management strategy was guided by clinical symptoms, tumor characteristics such as localization, size, and growth rate, and patient characteristics such as age, comorbidity, and patient preferences. A wait and scan policy, radiotherapy, or surgical resection were possible treatment options.

### 2.3. Mortality and Survival

For this study, follow-up data from *SDHB* and *SDHD* variant carriers were included from the date of the DNA test. In cases where clinical follow-up was available for the period before the DNA test, this period was not considered in the mortality analysis because it would have introduced immortal time bias [17]. Follow-up was defined as the time between the DNA test and the last clinical follow-up date before the end of the study period. Patients who were alive at the last clinical follow-up were classified as alive. Follow-up ended at the end of the study period, at the date of death or, in case of emigration, at the date of emigration [13]. To compare mortality between *SDHB* and *SDHD* variant carriers and the general population, the standardized mortality ratio (SMR) was estimated. Mortality rates for the Dutch population were obtained from Statistics Netherlands (CBS, The Netherlands) [18], using rates stratified by sex, age (per 1 year) and date (1-year periods). The SMR was calculated by dividing the observed number of deaths in the *SDHB* and *SDHD* cohorts. The expected number of deaths was calculated as the sum of the stratified number of expected deaths (stratum-specific mortality rates from the general population times follow-up time at risk). 

Survival was graphically displayed for *SDHB* and *SDHD* variant carriers by plotting survival in the carriers against the expected survival based on matched data from the general population. STATA 14.0 (Stata Corp, Texas, USA) was used for statistical analysis.

## 3. Results

In total, 192 *SDHB* variant carriers and 232 *SDHD* variant carriers were included in this study. The clinical characteristics are depicted in Table 1. The mean age at identification of the pathogenic gene variant was 46 years (range 9–77) in *SDHB* variant carriers and 44 years (range 16–73) in *SDHD* variant carriers. In total, 53 *SDHB* variant carriers (27.6%) and 198 *SDHD* variant carriers (85.3%) were diagnosed with HNPGL, either at time of presentation or during follow-up. Four *SDHB* patients (2.1%) and 16 *SDHD* patients (6.9%) developed PHEO and 26 *SDHB* patients (13.5%) and 18 (7.8%) *SDHD* patients developed sPGL. Malignant PGL, defined as metastatic PGL in non-paraganglionic tissue, were diagnosed in 14 *SDHB* (7.3%) and four *SDHD* patients (1.7%). Most *SDHB* variant carriers (110/193; 57.3%) were not affected at the time of DNA testing or during follow-up. In contrast, the majority of *SDHD* variant carriers was diagnosed with *SDHD*-associated disease (203/232; 87.5%). Details of the specific *SDHB* and *SDHD* variants are included in Appendix A. 

### Mortality and SMR 

Mortality data were available for all *SDHB* and *SDHD* variant carriers. The mean follow-up period was 3.0 (range 0–14.5) and 5.1 (range 0–12.5) years, respectively, for *SDHB* and *SDHD* variant carriers. In total, 6/192 (3.1%) *SDHB* variant carriers died at age 32, 37, 49, 52, 62, and 63. In three patients the cause of death was directly related to progressive PGL disease. In contrast, 5/232 (2.2%) *SDHD* variant carriers died at age 41, 43, 71, 71, and 74. In two cases the cause of death was most likely associated with PGL disease. Clinical characteristics of the variant carriers who died during the study period are listed in Table 2. 

A direct comparison between *SDHB* and *SDHD* variant carriers is hampered by the limited number of carriers and the heterogeneity between both groups. We performed an adjusted Poisson regression, adjusting for age, sex, and calendar time. The rate ratio comparing *SDHB* to *SDHD* variant carriers was 0.48 (95% confidence interval (CI) 0.15–1.62). However, the power for this analysis is low. As both groups have few events, we cannot draw conclusions from the non-significant *p*-value.

For the comparison of both the *SDHB*- and *SDHD*-linked cohorts with normative data of the Dutch population, a total of 1781 person-years were available (*SDHB* 590 and *SDHD* 1191 years, respectively). The SMR for *SDHB* mutation carriers was 1.89 (95% confidence interval (CI) 0.85–4.21) (Figure 1). A separate analysis including only symptomatic *SDHB* variant carriers—i.e., those with manifest disease—showed a higher SMR at 2.88 (95% CI 1.08–7.68). These results suggest an increased mortality risk for *SDHB* variant carriers compared to the general Dutch population, especially for carriers affected by *SDHB*-associated disease. For *SDHD* variant carriers, the SMR was 0.93 (95% CI 0.39–2.23), increasing only slightly to 1.06 (95% CI 0.44–2.54) in affected carriers, suggesting that mortality is not increased in *SDHD* variant carriers.

## 4. Discussion

In this study we estimated the mortality for *SDHB* and *SDHD* pathogenic variant carriers. Whereas the mortality for *SDHD* variant carriers is comparable with a matched cohort of the general Dutch population (SMR = 0.93), *SDHB* variant carriers show a higher mortality (SMR = 1.89, meaning a 1.89 times higher risk of death than the matched cohort of the general Dutch population). 

These mortality ratios should be interpreted with some caution. First, not all deaths in our cohort are directly attributable to PGL-linked disease. However, a comparison is made with the mortality of the general Dutch population. Therefore, eliminating other causes of death would be inappropriate. 

Second, even though the *SDHB* variant carriers represent a nationwide cohort, PGL is a rare disease and patient numbers are inevitably limited. As a result, the study estimates have broad confidence intervals. In addition, the follow-up of the start of this study is defined as the time of DNA testing and not PGL/PHEO diagnosis. As the genetic causes of hereditary PGL syndromes have been determined only recently, follow-up is relatively limited. However, the differences between *SDHB* and *SDHD* variant carriers are remarkable, all the more so when considering that *SDHD* variants are characterized by a high penetrance of PGL (88–100%), and *SDHB* variants by a much lower lifelong PGL risk (22–42%) [5,6,7,8,9,10]. In *SDHD* variant carriers, the occurrence of often multiple associated (HN)PGL seems to have no clear impact on survival [12]. In contrast, *SDHB* variant carriers seem to face increased mortality even though they are under more intensive surveillance and, in our study, have a shorter follow-up. This decreased survival of *SDHB* variant carriers is attributable to the higher mortality of affected *SDHB* patients (SMR = 2.88). Moreover, the majority of deceased *SDHB*-linked patients suffered from progressive malignant PGL (Table 2). Unaffected *SDHB* variant carriers have a mortality ratio that is more in line with the general Dutch population (SMR = 1.12). 

It is intriguing that the causative gene seems to determine variation in the prognoses for PGL/PHEO patients, even though pathogenic variants in *SDHB* and *SDHD* cause PGL/PHEO syndrome through defects in the same protein complex (succinate dehydrogenase, SDH). We speculate that this could be the result of intrinsic properties of the *SDHB*-associated PGL/PHEO syndrome, a deleterious effect of *SDHB* variants on other factors that influence survival, or differences between *SDHB* and *SDHD* variants in the potential to induce other types of malignancy. Interestingly, other types of malignancies (i.e. prostate cancer, lung cancer, breast cancer) are listed as causes of death both in the *SDHB*- and *SDHD*-linked cohorts (see Table 2). Although the *SDHx*-associated tumor spectrum is expanding, none of these malignancies have been directly linked to *SDHB* or *SDHD* variants. Even so, *SDHD* and/or *SDHB* variants could alter the susceptibility to certain types of malignancy other than PGL/PHEO. Indeed, 0.25% and 0.05% of breast cancer exomes carry somatic *SDHB* and *SDHD* variants, respectively [19,20].

The finding that all deceased *SDHB*-related PGL patients had metastatic PGL suggests that the occurrence of metastatic disease in *SDHB*-linked PGL syndrome particularly impacts survival, and that metastases may be either more prevalent in *SDHB*-linked cases, as suggested before [7,10,21,22,23,24], or more aggressive than metastatic diseases associated with other *SDHx* genes, a finding that is in line with the very poor 5-year survival rate of *SDHB*-linked metastatic disease reported by Amar et al. [11]. Another explanation might be that metastases from sPGL behave more aggressively than those of parasympathetic HNPGL, and that these sPGL are more prevalent in *SDHB*-linked disease [13,25]. Indeed, the PGL patients that died of progressive PGL disease both in the *SDHB*- and *SDHD*-linked cohorts all suffered from primary sPGL tumors.

The difference in the mortality between *SDHB* and *SDHD* variant carriers is another clear indication that causative genetic alteration is of critical importance to the outcome and risks of an individual PGL patient. This is important in counseling PGL/PHEO patients, but may also warrant gene-specific management strategies for PGL patients. In the present study, however, we did not evaluate the effect of PGL follow-up protocols or treatment on survival. From the patients that died of *SDHB*-related disease (*n* = 3), two already had proven metastatic disease at the time of diagnosis. Surgical resection with tumor-free margins seems to be a logical treatment strategy when trying to avoid progression of the disease, but there may be undetected metastases already present at the time of surgery [26,27]. The observation that the higher mortality associated with *SDHB* variant carriers seems to be attributable to patients that are affected by metastatic sPGL may warrant a more aggressive surgical strategy towards sPGL tumors in *SDHB*-linked patients. The risk of the malignant transformation of an sPGL tumor left untreated is, however, unknown. This unknown risk of disease progression must be weighed against the risk of surgical morbidity [28].

## 5. Conclusion

In conclusion, compared to a matched cohort of the general population, mortality is increased in *SDHB* variant carriers but not in *SDHD* variant carriers. This insight emphasizes the significance of DNA-testing; gene-specific clinical risks may warrant tailored management strategies. Further research is necessary to demonstrate the effect of (early) intervention of PGL/PHEO on mortality rates, especially in *SDHB* variant carriers. 

## Figures and Tables

**Figure 1 cancers-11-00103-f001:**
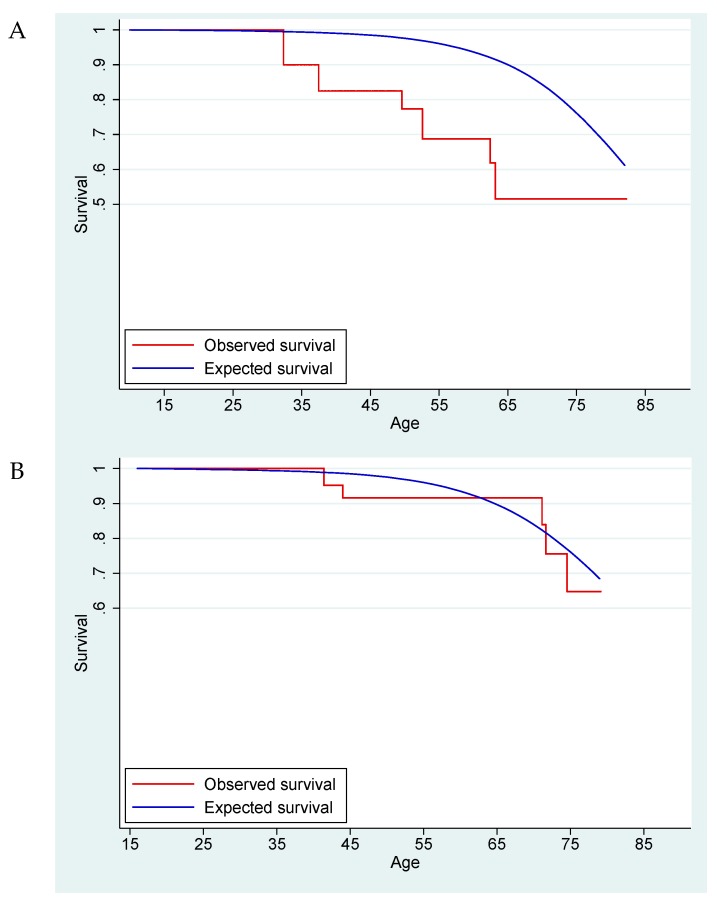
The Kaplan–Meier survival curve for *SDHB* variant carriers (**A**) and *SDHD* variant carriers (**B**) compared with the expected survival based on the general Dutch population.

**Table 1 cancers-11-00103-t001:** Clinical characteristics of carriers of pathogenic variants in succinate dehydrogenase subunits B and D (*SDHB* and *SDHD*).

Clinical Characteristics	*SDHB* *n* = 192	*SDHD* *n* = 232
Male (%)/female (%)	81 (42.2)/111 (57.8)	123 (53.0)/109 (47.0)
Mean age at genetic testing	46 years (range 9–77)	44 years (range 16–73)
HNPGL (%)	53 (27.6)	198 (85.3)
sPGL (%)	26 (13.5)	18 (7.8)
Pheochromocytoma (%)	4 (2.1)	16 (6.9)
Malignant PGL (%)	14 (7.3)	4 (1.7)
Unaffected (%)	110 (57.3)	30 (12.9)

HNPGL = head and neck paraganglioma, sPGL = sympathetic paraganglioma, PGL = paraganglioma.

**Table 2 cancers-11-00103-t002:** Details of six *SDHB* and five *SDHD* variant carriers who died during follow-up.

Sex	Mutation	Predicted Protein Change	Location of PGL	Age at PGL Diagnosis (years)	Age at Diagnosis of Malignant Disease (years)	Age at Death (years)	Location of Metastases	Cause of Death
*M*	*SDHB* exon 3 deletion	p.?	Presacral	28	28	32	Bone	Progressive malignant PGL
*F*	*SDHB* c.654G > A	p.(Trp218*)	Bladder	19	58	62	Lymph nodes, bone	Progressive malignant PGL
*F*	*SDHB* exon 3 deletion	p.?	Para-vertebral abdominal	33	33	37	Lymph nodes, bone	Progressive malignant PGL
*F*	*SDHB* c.727T > A	p.(Cys243Ser)	Retroperitoneal (para-aortic)	52	55	63	Bone	Myocardial infarction, heart failure and acute respiratory distress syndrome
*F*	*SDHB* c.423 + 1G > A	p.?	n.a.	49	n.a.	52	n.a.	Respiratory insufficiency due to lung bleeding after chemoradiotherapy for lung cancer
*F*	*SDHB* c.423 + 1G > A	p.?	n.a.	42	n.a.	49	n.a.	Metastatic breast cancer
F	*SDHD* c.274G > T	p.(Asp92Tyr)	Bladder	42	42	43	Lymph nodes, bone marrow	Progressive malignant PGL
F	*SDHD* c.274G > T	p.(Asp92Tyr)	Mediastinal	67	67	74	Lymph nodes, bone	Unknown, however the patient was known to have progressive malignant PGL
F	*SDHD* c.274G > T	p.(Asp92Tyr)	Bilateral CBT, VBT	55	n.a.	71	n.a.	Cardiac arrest
F	*SDHD* c.242C > T	p.(Pro81Leu)	CBT	38	n.a.	41	n.a.	Breast cancer
M	*SDHD* c.274G > T	p.(Asp92Tyr)	CBT, jugular PGL, retroperitoneal	52	n.a.	71	n.a.	Prostate cancer

PGL = paraganglioma, CBT = carotid body tumor, VBT = vagal body tumor, n.a. = not applicable.

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
