# Peer review of "Increased Mortality in SDHB but Not in SDHD Pathogenic Variant Carriers"

_cancers, 2019, doi:10.3390/cancers11010103_

Round 1
Reviewer 1 Report
The authors present a large Dutch cohort of SDHB and SDHD pathogenic variant carriers and compare mortality standard mortality ratios. It is a noteworthy study, but its presentation could be improved.
The major limitation in this study is the short time period of follow-up period of 3-5 years and the modest number of patients who died. This may be a limit which in the interest of time for publication, is not feasible to extend. It should be mentioned in the discussion. Also, is the difference in the SMRs between SDHB and SDHD statistically different?
Another approach to the analysis is to compare the kaplan-meier curves of SDHB vs SDHD (instead of expected). Is there a statistically significant difference?
Firstly, there are a few areas of the study which should be further emphasized. The fact that more SDHD carriers presented with paragangliomas vs SDHB carriers, demonstrates that there is less bias towards SDHD carriers. Despite this SMR was less in SDHD carriers.
Secondly, more intensive surveillance was offered to SDHB carriers, and despite this, SMR was higher in these individuals.
Thirdly, despite the shorter follow-up for SDHB carriers, they had increased mortality.
It would be interesting to compare and contrast the types of mutations seen in the SDHB (most are stop-gains/splice) vs missense in the SDHDs. To this end, the p. should be added to Table 2 for the reader to see easily the effect of the mutation.
The authors could also hypothesize why SDHB carriers had negative outcomes in the two cases that did not have PGL and how SDHB may affect other aspects of disease.
There are some typographical errors which need addressing. A few include:
-The authors should consistently use pathogenic variant (vs variant) eg Line 69.
-The majority of SDHB [pathogenic] variants carriers was (or were?)
-therefore line 110
Also the overall format of the paper is awkward and may be a reflection of the journal's requirements. The introduction, results, and discussion is folloed by the subjects and methods and conclusions. Introduction, subjects and methods, results, discussion, and finally conclusions would have been easier to read, as it was not clear what type of surveillance patients had and their characteristics.
Author Response
Manuscript ID: cancers-419269, revised version
Title: Increased mortality in SDHB but not in SDHD pathogenic variant carriers
Authors: J.A.Rijken et al.
We thank the reviewers for their careful comments. Below we comment on the issues raised by the reviewers. We have highlighted the revisions using the ‘Track Changes’ function in Word. We believe that we have addressed all issues in our revised manuscript. Please find below our reaction to the comments of the reviewers.
Reviewers’ comments to the author:
Reviewer 1
The major limitation in this study is the short time period of follow-up period of 3-5 years and the modest number of patients who died. This may be a limit which in the interest of time for publication, is not feasible to extend. It should be mentioned in the discussion.
We thank the reviewer for this comment. We have highlighted the limitations of the study in the Discussion section:
“…even though the SDHB variant carriers represent a nationwide cohort, PGL is a rare disease and therefore patients numbers are inevitably limited. As a result the study estimates have broad confidence intervals. In addition, the follow up of the start of this study is defined as the time of DNA testing and not PGL/PHEO diagnosis. As the genetic causes of hereditary PGL syndromes have been elucidated relatively recently, follow up is also relatively limited…”
Also, is the difference in the SMRs between SDHB and SDHD statistically different?
Another approach to the analysis is to compare the kaplan-meier curves of SDHB vs SDHD (instead of expected). Is there a statistically significant difference?
In this study we estimated the SMR for SDHB and SDHD pathogenic variant carriers and compared both SMR with the SMR of a matched cohort of the general Dutch population.
We agree that a direct comparison between SDHD and SDHB is of interest.
A direct comparison between SDHB and SDHD variant carriers is hampered by the limited number of carriers and the heterogeneity between both groups.
We did not perform a Kaplan Meier analysis, as this would not allow us to adjust for age, sex and calendar time. We however performed an adjusted Poisson regression. The rate ratio (adjusted for age, sex and calandar time) comparing SDHD to SDHB carries was 0.48 (95% CI 0.15-1.62). However, the power for this analysis is even lower as now both groups have few events; so we cannot draw conclusions from the non-significant p-value.
We have added:
“…A direct comparison between SDHB and SDHD variant carriers is hampered by the limited number of carriers and the heterogeneity between both groups. We performed an adjusted Poisson regression, adjusting for age, sex and calendar time. The rate ratio comparing SDHB to SDHD variant carriers was 0.48 (95% CI 0.15-1.62), however the power for this analysis is low. As both groups have few events we cannot draw conclusions from the non-significant p-value.…”
to the Results section.
Firstly, there are a few areas of the study which should be further emphasized. The fact that more SDHD carriers presented with paragangliomas vs SDHB carriers, demonstrates that there is less bias towards SDHD carriers. Despite this SMR was less in SDHD carriers.
Secondly, more intensive surveillance was offered to SDHB carriers, and despite this, SMR was higher in these individuals.
Thirdly, despite the shorter follow-up for SDHB carriers, they had increased mortality.
We have added:
“…In contrast, SDHB variant carriers seem to have an increased mortality even though they are under more intensive surveillance and, in our study, have a shorter follow up…”
to the Discussion section.
It would be interesting to compare and contrast the types of mutations seen in the SDHB (most are stop-gains/splice) vs missense in the SDHDs. To this end, the p. should be added to Table 2 for the reader to see easily the effect of the mutation.
We have added the column ‘Predicted protein change’ in Table 2. The predicted protein change of all SDHD and SDHB variant carriers encounted in this study are listed in Appendix 1.
The authors could also hypothesize why SDHB carriers had negative outcomes in the two cases that did not have PGL and how SDHB may affect other aspects of disease.
We agree with the reviewer that other aspects of disease linked to SDHB pathogenic gene variants are of interest. We have added the following to the Discussion section (including two extra refs):
We speculate that this could be the result of intrinsic properties of the SDHB-associated PGL/PHEO syndrome, a deleterious effect of SDHB variants on other factors that influence survival, or differences between SDHB and SDHD variants in the potential to induce other types of malignancy. Interestingly, other types of malignancies (i.e. prostate cancer, lung cancer, breast cancer) are listed as cause of death both in the SDHB and SDHD-linked cohorts (see Table 2). Although the SDHx-associated tumor spectrum is expanding, none of these malignancies have been directly linked to SDHB or SDHD variants. Even so, SDHD and/or SDHB variants could alter the susceptibility to certain types of malignancy other than PGL/PHEO. Indeed, 0.25% and 0.05% of breast cancer exomes carry somatic SDHB and SDHD variants respectively [19,20].
19. John G Tate, Sally Bamford, Harry C Jubb, Zbyslaw Sondka, David M Beare, Nidhi Bindal, Harry Boutselakis, Charlotte G Cole, Celestino Creatore, Elisabeth Dawson, Peter Fish, Bhavana Harsha, Charlie Hathaway, Steve C Jupe, Chai Yin Kok, Kate Noble, Laura Ponting, Christopher C Ramshaw, Claire E Rye, Helen E Speedy, Ray Stefancsik, Sam L Thompson, Shicai Wang, Sari Ward, Peter J Campbell, Simon A Forbes; COSMIC: the Catalogue Of Somatic Mutations In Cancer, Nucleic Acids Research, Volume 47, Issue D1, 8 January 2019, Pages D941–D947, https://doi.org/10.1093/nar/gky1015
20. Oudijk L, Gaal J, de Krijger RR. The Role of Immunohistochemistry and Molecular Analysis of Succinate Dehydrogenase in the Diagnosis of Endocrine and Non-Endocrine Tumors and Related Syndromes. Endocr Pathol. 2018 Nov 12. doi: 10.1007/s12022-018-9555-2.
There are some typographical errors which need addressing. A few include:
-The authors should consistently use pathogenic variant (vs variant) eg Line 69.
We agree with the reviewer that consistent use of terminology is important. In order to avoid unnecessary repeats of the word pathogenic we have added…
‘…The cohort of pathogenic germline variant carriers (hereafter variants)…’
in the Subjects and Methods section.
-The majority of SDHB [pathogenic] variants carriers was (or were?)
-therefore line 110
We have corrected these errors.
Also the overall format of the paper is awkward and may be a reflection of the journal's requirements. The introduction, results, and discussion is folloed by the subjects and methods and conclusions. Introduction, subjects and methods, results, discussion, and finally conclusions would have been easier to read, as it was not clear what type of surveillance patients had and their characteristics.
We agree with the reviewer, however the format of the manuscript was adjusted to the journal requirements.

Reviewer 2 Report
This study by Rijken and co-workers is an effort to estimate the overall mortality in nation-wide cohorts with established germline mutations in SDHB and SDHD. These numbers were then compared to the general populace. Mortality was found increased in SDHB variant carriers, especially in those affected by PGL. In SDHD variant carriers however, the mortality was comparable to the numbers in the general population. The work is simple in design, yet impactful and of interest to the general clinical audience. The authors address potential pitfalls to their analyses, and the conclusions match the obtained results. This referee only has minor comments:
The definition of a germline mutation in this study was stated as: "In this manuscript we report pathogenic or probably pathogenic variants, including missense mutations in highly conserved regions that are probably pathogenic, as germline mutations" - rows 169-171. However, I cannot find any in silico mutational prediction data for the variants presented in Appendix 1. I therefore wonder how the authors came up with the conclusion that these variants are pathogenic - not least the intronic variants. I know these cohorts have been published previously, but I would like to have the overall methodology (mutation prediction, aa conservation analyses or similar) described in brief in the current article rather than to simply refer to previous studies.
Lines 120-122: "Moreover, the majority of deceased SDHB-linked patients died of the consequences of progressive malignant PGL (Table 2)." To me, the table show that "only" 3/6 SDHB carriers (50%) died of progressive PGL disease. Please clarify.
Several patients with germline mutations of either gene also presented with other cancer types (lung cancer, breast cancer, prostate cancer). It seems as though two patients died of breast cancer at ages 41 and 49 respectively - which are fairly young ages for fatal breast cancer, and especially interesting with two cases in a combined cohort of 11 cases (18%). I would like to see a short paragraph in the Discussion section in which the coupling (or lack of coupling) between SDHB/D and other cancer types are discussed - as an eventual known overrepresentation of various cancer forms might affect the mortality beyond the PGL-associated mortality. (Indeed, if you consult the COSMIC database, 0.25% and 0.05% of breast cancer exomes carry somatic SDHB and SDHD mutations respectively.)
Author Response
Manuscript ID: cancers-419269, revised version
Title: Increased mortality in SDHB but not in SDHD pathogenic variant carriers
Authors: J.A.Rijken et al.
We thank the reviewers for their careful comments. Below we comment on the issues raised by the reviewers. We have highlighted the revisions using the ‘Track Changes’ function in Word. We believe that we have addressed all issues in our revised manuscript. Please find below our reaction to the comments of the reviewers.
Reviewers’ comments to the author:
Reviewer 2
This study by Rijken and co-workers is an effort to estimate the overall mortality in nation-wide cohorts with established germline mutations in SDHB and SDHD. These numbers were then compared to the general populace. Mortality was found increased in SDHB variant carriers, especially in those affected by PGL. In SDHD variant carriers however, the mortality was comparable to the numbers in the general population. The work is simple in design, yet impactful and of interest to the general clinical audience. The authors address potential pitfalls to their analyses, and the conclusions match the obtained results. This referee only has minor comments:
The definition of a germline mutation in this study was stated as: "In this manuscript we report pathogenic or probably pathogenic variants, including missense mutations in highly conserved regions that are probably pathogenic, as germline mutations" - rows 169-171. However, I cannot find any in silico mutational prediction data for the variants presented in Appendix 1. I therefore wonder how the authors came up with the conclusion that these variants are pathogenic - not least the intronic variants. I know these cohorts have been published previously, but I would like to have the overall methodology (mutation prediction, aa conservation analyses or similar) described in brief in the current article rather than to simply refer to previous studies.
We thank the reviewer for this comment and have adjusted this paragraph in the Subjects and methods section:
‘…In this manuscript we report pathogenic or probably pathogenic variants, including missense mutations in highly conserved regions that are probably pathogenic partially based on mutation prediction analyses, as germline mutations. Information on amino acid conservation can be found in the LOVD database [http://chromium.liacs.nl/lovd_sdh], further information including mutation prediction analyses can be obtained on request…’
Lines 120-122: "Moreover, the majority of deceased SDHB-linked patients died of the consequences of progressive malignant PGL (Table 2)." To me, the table show that "only" 3/6 SDHB carriers (50%) died of progressive PGL disease. Please clarify.
We agree with the reviewer that in 3/6 of SDHB-linked cases the cause of death was directly related to progressive malignant PGL. In the fourth case the relation between cause of death and malignant PGL was suggestive, but not as clear-cut. We have altered the text accordingly:
‘…Moreover, the majority of deceased SDHB-linked patients suffered from progressive malignant PGL (Table 2)…’
Several patients with germline mutations of either gene also presented with other cancer types (lung cancer, breast cancer, prostate cancer). It seems as though two patients died of breast cancer at ages 41 and 49 respectively - which are fairly young ages for fatal breast cancer, and especially interesting with two cases in a combined cohort of 11 cases (18%). I would like to see a short paragraph in the Discussion section in which the coupling (or lack of coupling) between SDHB/D and other cancer types are discussed - as an eventual known overrepresentation of various cancer forms might affect the mortality beyond the PGL-associated mortality. (Indeed, if you consult the COSMIC database, 0.25% and 0.05% of breast cancer exomes carry somatic SDHB and SDHD mutations respectively.)
We agree with the reviewer that other types of cancer related to SDH genes are of interest in this respect. We have added the following to the discussion section (including two extra refs):
‘ … We speculate that this could be the result of intrinsic properties of the SDHB-associated PGL/PHEO syndrome, a deleterious effect of SDHB variants on other factors that influence survival, or differences between SDHB and SDHD variants in the potential to induce other types of malignancy. Interestingly, other types of malignancies (i.e. prostate cancer, lung cancer, breast cancer) are listed as cause of death both in the SDHB and SDHD-linked cohorts (see Table 2). Although the SDHx-associated tumor spectrum is expanding, none of these malignancies have been directly linked to SDHB or SDHD variants. Even so, SDHD and/or SDHB variants could alter the susceptibility to certain types of malignancy other than PGL/PHEO. Indeed, 0.25% and 0.05% of breast cancer exomes carry somatic SDHB and SDHD variants respectively [19,20].…’
19. John G Tate, Sally Bamford, Harry C Jubb, Zbyslaw Sondka, David M Beare, Nidhi Bindal, Harry Boutselakis, Charlotte G Cole, Celestino Creatore, Elisabeth Dawson, Peter Fish, Bhavana Harsha, Charlie Hathaway, Steve C Jupe, Chai Yin Kok, Kate Noble, Laura Ponting, Christopher C Ramshaw, Claire E Rye, Helen E Speedy, Ray Stefancsik, Sam L Thompson, Shicai Wang, Sari Ward, Peter J Campbell, Simon A Forbes; COSMIC: the Catalogue Of Somatic Mutations In Cancer, Nucleic Acids Research, Volume 47, Issue D1, 8 January 2019, Pages D941–D947, https://doi.org/10.1093/nar/gky1015
20. Oudijk L, Gaal J, de Krijger RR. The Role of Immunohistochemistry and Molecular Analysis of Succinate Dehydrogenase in the Diagnosis of Endocrine and Non-Endocrine Tumors and Related Syndromes. Endocr Pathol. 2018 Nov 12. doi: 10.1007/s12022-018-9555-2.
